# Mesenchymal Stromal Cells Primed by Toll-like Receptors 3 and 4 Enhanced Anti-Inflammatory Effects against LPS-Induced Macrophages via Extracellular Vesicles

**DOI:** 10.3390/ijms242216264

**Published:** 2023-11-13

**Authors:** Sein Hwang, Dong Kyung Sung, Young Eun Kim, Misun Yang, So Yoon Ahn, Se In Sung, Yun Sil Chang

**Affiliations:** 1Department of Health Sciences and Technology, SAIHST, Sungkyunkwan University, Seoul 06351, Republic of Korea; seinh4007@g.skku.edu; 2Cell and Gene Therapy Institute, Samsung Medical Center, Seoul 06351, Republic of Korea; dbible@skku.edu (D.K.S.); duddms920@skku.edu (Y.E.K.); misun.yang@samsung.com (M.Y.); soyoon.ahn@samsung.com (S.Y.A.); sein.sung@samsung.com (S.I.S.); 3Department of Pediatrics, Samsung Medical Center, Sungkyunkwan University School of Medicine, Seoul 06351, Republic of Korea

**Keywords:** mesenchymal stromal cell

## Abstract

Although it has been suggested that toll-like receptor (TLR) 3 and TLR4 activation alters mesenchymal stromal cells (MSCs)’ immunoregulatory function as anti- or pro-inflammatory phenotypes, we have previously confirmed that TLR4-primed hUCB-MSCs alleviate lung inflammation and tissue injury in an *E. coli*-induced acute lung injury (ALI) mouse model. Therefore, we hypothesized that strong stimulation of TLR3 or TLR4 prompts hUCB-MSCs to exhibit an anti-inflammatory phenotype mediated by extracellular vesicles (EVs). In this study, we compared the anti-inflammatory effect of TLR3-primed and TLR4-primed hUCB-MSCs against an LPS-induced ALI in vitro model by treating MSCs, MSC-derived conditioned medium (CM), and MSC-derived extracellular vesicles (EVs). LPS-induced rat primary alveolar macrophage and RAW 264.7 cells were treated with naïve, TLR3-, and TLR4-primed MSCs and their derived CM and EVs. Flow cytometry and ELISA were used to evaluate M1-M2 polarization of macrophages and pro-inflammatory cytokine levels, respectively. LPS-stimulated macrophages showed significantly increased pro-inflammatory cytokines compared to those of the normal control, and the percentage of M2 macrophage phenotype was predominantly low. In reducing the inflammatory cytokines and enhancing M2 polarization, TLR3- and TLR4-primed MSCs were significantly more effective than the naïve MSCs, and this finding was also observed with the treatment of MSC-derived CMs and EVs. No significant difference between the efficacy of TLR3- and TLR-primed MSCs was observed. Strong stimulation of TLR3- and TLR4-stimulated hUCB-MSCs significantly reduced pro-inflammatory cytokine secretion from LPS-induced macrophages and significantly enhanced the M2 polarization of macrophages. We further confirmed that TLR-primed MSC-derived EVs can exert anti-inflammatory and immunosuppressive effects alone comparable to MSC treatment. We hereby suggest that in the LPS-induced macrophage in vitro model, EVs derived from both TLR3 and TLR4-primed MSCs can be a therapeutic candidate by promoting the M2 phenotype.

## 1. Introduction

Mesenchymal stromal cells (MSCs) are multipotent stromal cells that have been well recognized for their potential in cell therapies in multiple disease models, due to their regenerative and immunomodulatory effect [1,2,3]. MSCs can be isolated from numerous sources, such as bone marrow, adipose tissue, and umbilical cord blood (UCB). UCB-derived MSCs are good candidates for cell therapeutics owing to their high proliferation capacity, anti-inflammatory properties, noninvasive procedures, and fewer ethical issues [4,5]. The safety and efficacy of UCB-derived MSC therapy have been tested in numerous studies, including animal studies on bacterial acute lung injury and meningitis as well as in clinical trials for neonatal bronchopulmonary dysplasia and intraventricular hemorrhage [6,7,8,9,10].

The anti-inflammatory response is essential for MSCs to attenuate cellular and tissue injuries, and various priming methods have been explored to boost their anti-inflammatory functions in the host [11,12,13]. In innate immunity, toll-like receptors (TLRs) are pattern recognition receptors that mediate immune responses by recognizing molecular patterns such as damage-associated molecular patterns (DAMP) and pathogen-associated molecular patterns (PAMP). Among the multiple TLR family members, TLR3 recognizes nucleic acids released after host cellular damage or exogenous viral infection, whereas TLR4 recognizes lipopolysaccharide (LPS) molecules associated with Gram-negative bacteria [14,15,16,17]. There are controversial debates on the changes in the immunoregulatory function of MSCs upon TLR4 activation and whether it shifts MSCs to a pro-inflammatory or anti-inflammatory phenotype [18,19,20,21,22,23,24,25]. We previously confirmed that TLR4-primed hUCB-MSCs significantly enhanced the M2-type alveolar macrophage polarization and attenuated lung inflammation and tissue injury in an *E. coli*-induced acute lung injury mouse model [26]. Bacterial infection is one of the most common causes of ALI [27,28]. During infection and subsequent inflammation in ALI, LPS derived from Gram-negative bacteria and nucleic acids released from necrotic cells, referred to as PAMPs and DAMPs, respectively, have the potential to trigger TLR4 and TLR3 activation [14,15,16,17,29,30,31,32]. Consequently, in a mouse model, upon *E. coli* induction, pro-inflammatory cytokines such as Il-1α, IL-1β, IL-6, and TNF-α are significantly increased, while anti-inflammatory cytokines such as IL-4 and IL-10 are decreased [26]. Our hypothesis was that in the presence of an abundance of PAMP and DAMP signals or ligands for TLR3 and TLR4, transplanted MSCs would exhibit a reparative response to counteract the damage caused by ongoing inflammation. This assumption was based on the fact that both TLR3 and TLR4 share a pathway dependent on a TIR domain-containing adaptor protein inducing an interferon-beta (TRIF)-dependent pathway [33,34,35]. Since we previously confirmed the therapeutic efficacy of TLR4-primed hUCB-MSCs transplanted into a mouse model of *E. coli*-induced ALI, this study aimed to validate the corresponding findings in vitro by exposing macrophages to TLR3- and TLR4-primed hUCB-MSCs under LPS-induced active inflammation [9,26]. We hypothesized that upon strong stimulation with TLR3 and TLR4, hUCB-MSCs would respond with an anti-inflammatory phenotype and that their immunomodulatory effects would be mediated by extracellular vesicles (EVs).

## 2. Results

### 2.1. Increasing Doses of Poly(I:C) and LPS Increased the Secretion of Pro-Inflammatory Cytokine and Growth Factors without Changes in the Viability of hUCB-MSCs

Poly(I:C) and LPS were used to prime TLR3 and TLR4, respectively, in hUCB-MSCs (MSCs). Incremental doses of poly (I:C) (100 ng, 1 µg, 10 µg, and 100 µg) and LPS (10 ng, 100 ng, 1 µg, and 10 µg) were used to investigate their effect on the viability and production of cytokines and growth factors (Figure 1).

No significant cell death was observed at any dose of poly (I:C) or LPS. Upon stimulation with all poly (I: C) doses, hUCB-MSCs secreted significantly higher levels of IL-6 than the normal control group (NC). Stimulation with 1 µg, 10 µg, and 100 µg of poly(I:C) caused significantly higher secretion of IL-6 compared to stimulation with 100 ng of poly(I:C), and stimulation with 10 µg and 100 µg resulted in higher levels than stimulation with 1 µg of poly(I:C). No significant difference was noted in TNF-α levels upon stimulation with any doses of poly(I:C). Stimulation with 10 µg and 100 µg of poly(I:C) induced significantly higher levels of VEGF and HGF compared to the NC group, while lower doses did not. Stimulation with 100ng, 1 µg, and 10 µg of LPS induced significantly higher levels of IL-6 compared to the NC group. Moreover, the IL-6 levels induced by 100 ng, 1 µg, and 10 µg of LPS were significantly greater than those with 10 ng of LPS. No significant difference was observed in TNF-α levels upon stimulation at all doses of LPS. Stimulation with 100 ng, 1 µg, and 10 µg of LPS induced significantly higher levels of VEGF compared to those observed in the NC group. Stimulation with 10 ng, 100 ng, 1 µg, and 10 µg of LPS induced significantly higher levels of HGF compared to those observed in the NC group. Doses of 100 ng, 1 µg, and 10 µg of LPS resulted in greater HGF stimulation than 10 ng of LPS.

### 2.2. Effect of TLR3- and TLR4-Primed MSCs in Cytokine Secretion by LPS-Induced Alveolar Macrophages

LPS stimulation significantly increased the levels of pro-inflammatory cytokines IL-6, TNF-a, and IL-1a compared to the NC group in RAW264.7 cells, a mouse alveolar macrophage cell line (Figure 2A). LPS-induced RAW264.7 cells were either untreated or treated with naïve MSCs, TLR3-primed MSCs, or TLR4-primed MSCs. The naïve MSC-treated group showed a significant reduction in inflammatory cytokine levels of IL-6, TNF-a, and IL-1a compared to the NC group. However, further significant reductions in these inflammatory cytokine levels were observed in the TLR3- and TLR-4-primed MSCs compared to those in the naïve MSC-treated group. No significant difference was observed in IL-6, TNF-a, and IL-1a levels between TLR3- and TLR4-primed MSC-treated groups. Consistent results were obtained in an LPS-induced primary cultured rat alveolar macrophage (AM) model, as an ex vivo approach (Figure 2B). After the LPS induction, rat AM secreted significantly higher IL-6, TNF-a, and IL-1a levels compared to the NC group. The increase in inflammatory cytokine levels was significantly reduced after treatment with naive MSCs. However, after treatment with TLR3- and TLR4-primed MSCs, inflammatory cytokine levels were further reduced compared to those after naïve MSC treatment. Inflammatory cytokine levels were not significantly different between the TLR3- and TLR4-primed MSC-treated groups in the LPS-induced rat AM model.

### 2.3. Effect of TLR3- and TLR4-Primed MSC in M1 and M2 Polarization of LPS-Induced Alveolar Macrophage

The effect of MSCs on macrophage polarization in LPS-induced rat AM is shown in Figure 3. The extent of M2 polarization after LPS induction and treatment with naïve, TLR3-, and TLR4-primed MSCs was measured by FACS. The extent of M2 polarization in LPS-induced rat AM was only approximately 0.39%. When treated with naïve MSCs, the percentage of M2 phenotype increased significantly to approximately 27.51%. In the TLR3- and TLR4-primed MSC-treated groups, the M2 percentage significantly increased to 49.46% and 40.85%, respectively, compared to that in the naïve MSC-treated group. This suggests that TLR3- and TLR4-priming enhances the anti-inflammatory effect of MSCs against LPS-induced inflammation by modulating the macrophage phenotype.

### 2.4. Effect of Conditioned Medium and Extracellular Vesicles Derived from TLR3- and TLR4-Primed MSCs on LPS-Induced RAW 264.7 Cells

The immunomodulatory effect of MSCs is mediated by paracrine signaling, in which effective molecules are secreted. In the LPS-induced RAW264.7 cell model, we treated conditioned media (CMs) from naïve and TLR3- and TLR4-primed hUCB-MSCs to confirm the paracrine action (Figure 4).

Compared to the NC group, LPS-induced RAW 264.7 cells showed significantly increased levels of proinflammatory cytokines IL-6, TNF-a, and IL-1a. The LPS-induced increase in proinflammatory cytokine levels was significantly reduced after treatment with naïve MSC-CM. However, the effect of MSC-CM on reducing the inflammatory cytokine levels was significantly greater in TLR3 and TLR4-primed MSC-CM compared to naïve MSC-CM. Next, to investigate whether the EVs were responsible for the effects of CM, Es and EV-removed CMs from each MSC were independently treated with LPS-induced rat AM (Figure 5).

When the extent of macrophage polarization was quantified, in line with the results from MSC CM treatment, treatment with TLR3- and TLR4-primed MSC-derived EVs, compared to naive MSC-derived EVs, significantly increased the percentage of M2-type AM in the LPS-induced rat AM model. This effect was abolished when EV-removed CM from each MSC was treated. Again, no significant difference was noted between TLR3- and TLR4-primed MSCs in reducing the extent of LPS-induced inflammation or the modulatory effects on the macrophage phenotype shift.

## 3. Discussion

In the present study, we independently stimulated TLR3 and TLR4 of hUCB-MSCs and demonstrated that both TLR3- and TLR4-primed hUCB-MSCs exhibited immunomodulatory effects, substantially reducing LPS-induced inflammation in alveolar macrophages. These immunomodulatory effects were mediated by EVs secreted from MSCs. Notably, no significant difference was observed in the immunomodulatory effects between the two MSC priming methods. MSCs express multiple types of TLRs, and the role of TLR signaling in the therapeutic effects of MSCs must not be overlooked [31]. Plasticity of MSCs in immunoregulation allows them to “sense” the microenvironment and respond accordingly by either enhancing or reducing their immunomodulatory function [36,37]. Since multiple TLRs are expressed on hUCB-MSCs, the ‘sensing’ of the environment and its plasticity in regulating immune cells and tissue-specific cells emphasize the importance of TLR priming in considering MSC therapeutics. The ability of MSCs to control the extent of inflammation is critical not only in ALI but also in other severe inflammatory diseases, such as osteoarthritis [38]. Multiple studies have shown that the immunoregulatory function of MSCs changes upon activation of TLR3 and TLR4 [18,19,20,21,22,23,24,25]. Previously in our in vivo study, we confirmed that the anti-inflammatory and immunomodulatory effects of TLR4-primed hUCB-MSCs were mediated by SOCS3 and defensin [9,26]. However, the immunoregulatory responses of MSCs to the activation of different TLRs, specifically TLR3 and TLR4, remain controversial. Differences in the outcomes of TLR3- and TLR4-primed MSCs are assumed to pertain to different sources and donors of MSCs, priming protocols, environments to which MSCs are exposed, and targets of investigation, such as T cells or macrophages.

Regarding the source variance of MSCs, Jafari et al. have confirmed that the immunosuppressive function of TLR3- and TLR4-primed MSCs differed by the source of the MSC, with olfactory mucosa-derived MSC being more responsive than the adipose tissue-derived MSC to the same stimulation [24]. Li et al. confirmed that donor variations in the same source of MSC cause different immunoregulatory effects on TLR3 and TLR4 stimulation [25]. Prasanna et al. emphasized different sources of MSCs responding differently to inflammation [39]. These studies help to understand the discrepancies between many studies using different sources and donors of MSCs.

Different priming protocols may explain the different outcomes of priming with TLR3 and TLR4. Waterman et al. presented the concepts of the pro-inflammatory phenotype MSC1 and anti-inflammatory phenotype MSC2, also referred to as TLR4-primed MSC and TLR3-primed MSC, respectively [19]. However, in our previous studies, we confirmed the anti-inflammatory function of TLR4-primed MSCs in an ALI experimental model, similar to the present study. Studies consistent with ours state no difference between TLR3- and TLR4-primed MSCs [40,41], while those that have contrasting findings to our results state that TLR3-primed MSCs show an anti-inflammatory phenotype, whereas TLR4-primed MSCs show a pro-inflammatory phenotype [19,21,22,23]. The variation between these studies can be attributed to milder and shorter TLR3 and TLR4 stimulation of MSCs, different MSC sources of MSCs, and different immunomodulatory assessments, compared to the current study. Waterman et al. stimulated TLRs of hMSCs with LPS (10 ng/mL) and poly(I:C) (1 µg/mL) for 1 h and co-cultured TLR-primed MSCs with preactivated T cells for 72 h to measure the change in the percentage activation of T cells. Vega-Letter et al. similarly primed TLR3 and TLR4 of murine BM-MSCs with LPS (500 ng/mL) or poly(I:C) (10 µg/mL) for 1 h and co-cultured them with pre-activated T cells to assess T cell proliferation and differentiation into pro-inflammatory Th1 and Th17 [22]. In contrast to these studies but in parallel to the present study, Nemeth et al. confirmed that stimulation of BM-MSCs with a relatively high dose of LPS (10 µg) induced the secretion of significantly higher levels of IL-10, a major anti-inflammatory cytokine [42]. In the present study, upon an incremental increase in doses of poly(I:C) and LPS to hUCB-MSCs, at 10 µg/mL of poly(I:C) and 100 ng/mL of LPS, we observed a simultaneous steep increase in levels of IL-6, VEGF, and HGF (Figure 1). At higher doses (10 µg and 100 µg of poly(I:C); 100 ng, 1 µg, and 10 µg of LPS) where the pro-inflammatory cytokine IL-6 level was the highest, levels of VEGF and HGF were also highest. IL-6, a commonly known pro-inflammatory cytokine, is now recognized for its dual function as a pro- and anti-inflammatory cytokine. In addition to studies confirming IL-6-dependent macrophage M2 polarization, recent studies have emphasized the emerging role of MSC-derived IL-6 as a key regulator of immunomodulatory and therapeutic effects of MSCs [43,44,45,46,47]. In line with our data, Liu, Xu et al. confirmed that TNF-α and IFN-γ treated hUCB-MSCs induced significantly higher IL-6 and VEGF levels in hUCB-MSCs, which significantly induced M2 polarization of macrophages and enhanced therapeutic efficacy in wound healing [47]. Since high-dose stimulation with TLR3 and TLR4 significantly increased the secretion of IL-6, VEGF, and HGF, we hypothesized that a higher dose and longer stimulation with TLR3 and TLR4 would boost the anti-inflammatory and immunosuppressive ability of hUCB-MSCs. This concept of prolonged stimulation enhancing immunosuppressive ability was also suggested by Lin et al., who confirmed that stronger stimulation of MSCs drives M2 macrophage polarization, and Guryanova et al., who confirmed that 5-day pre-exposure to LPS protects mouse models from asthma-induced lung injury by modulating the macrophages [48,49]. The rationale behind this can be the endosomal degradation of LPS-TLR4 complex, reducing the amount of available TLRs that can initiate an inflammatory response, which in turn requires the new synthesis of TLRs for inflammation to be augmented. As such, the innate immune receptors undergo highly complex regulation, sensing both the amount of ligands and the duration of stimulation, resulting in a differential environment-dependent response. Therefore, understanding similar methods controlling the concentration and duration of receptor stimulation is critical when studying the effect of TLR stimulation.

Further differences in the outcomes of TLR3 and TLR4 activation of MSCs can be attributed to different experimental designs. In this study, we immediately co-cultured TLR-primed hUCB-MSCs with macrophages in the culture medium supplemented with LPS for 24 h. During the 24 h of LPS stimulation, significant levels of pro-inflammatory cytokines (IL-6, TNF-α, IL-1α) were secreted by macrophages, and the percentage of the M2 macrophage population was minimal (Figure 2 and Figure 3). When co-cultured with MSCs, both TLR3- and TLR4-primed hUCB-MSCs significantly reduced the cytokine levels and enhanced the M2 population more than the naïve hUCB-MSCs. There was no significant difference between TLR3- and TLR4-primed MSCs in both pro-inflammatory cytokine levels and M2 polarization of macrophages. TLR3- and TLR4-primed MSCs’ anti-inflammatory effects correlated with their ability to modulate macrophage M1-M2 polarization. Studies have shown that LPS and other inflammatory molecules stimulate MSCs to secrete immunosuppressive PGE2 or IDO, which polarize macrophages into the M2 phenotype, emphasizing the sensing and regulatory role of MSCs in the immune system [50,51,52]. TLR3 and TLR4 sharing the TRIF-dependent pathway could further explain the similar anti-inflammatory and immunomodulatory effects exerted by TLR3- and TLR4-primed MSCs [33,34,35]. While TLR3 only utilizes the TRIF-dependent pathway, TLR4 utilizes MyD88-dependent and TRIF-dependent pathways [31,53]. The TRIF-dependent pathway induces a type I IFN response, which leads to the activation of NF-κB, IRFs, and AP-1 [54]. We hypothesize that strong activation of TLR3 and TLR4 in hUCB-MSCs allowed convergence in the TRIF-dependent pathway, but further investigation is needed to confirm this.

To ensure the variation in the source of macrophages, we tested the anti-inflammatory effect of TLR3- and TLR4-primed hUCB-MSCs in both RAW264.7 cells and primary alveolar macrophages isolated from rat BAL fluid, and no significant difference was observed. The role of first-line resident macrophages is critical in bacterial ALI pathology because polarization towards the classical M1 phenotype can trigger excessive pro-inflammatory cytokine secretion, whereas the M2 phenotype can secrete anti-inflammatory cytokines and growth factors that enhance repair and regeneration [27,28,30,55]. Inflammatory bursts cause significant tissue damage and controlling M1-M2 polarization is critical for the pathophysiology and prognosis of ALI [29]. Since macrophages are key players in ALI pathology, we focused on TLR-primed MSCs’ immunomodulation of macrophages.

Moreover, we investigated whether TLR-primed MSC-derived EVs could modulate macrophages without direct MSC-macrophage cell contact with the MSC macrophages. Treatment of TLR3 and TLR4-primed hUCB-MSC-derived CMs and isolated EVs significantly reduced pro-inflammatory cytokines IL-6, TNF-α, and IL-1 α, and significantly enhanced M2 polarization of macrophages compared to the naïve hUCB-MSCs (Figure 4). There was no significant difference between the anti-inflammatory effects of TLR3- and TLR4-primed MSCs in both CM and EV treatment. EV-removed CMs abolished this result, confirming that the therapeutic effect of TLR3- and TLR4-primed hUCB-MSCs is mediated via EVs. MSCs’ therapeutic efficacy is known to be primarily mediated by a paracrine action through EVs [13,56]. Here, we not only confirmed the significance of EV in MSCs’ paracrine signaling but also that TLR-priming effects are mediated through EVs. Significantly increased VEGF levels in TLR3- and TLR4-primed hUCB-MSCs compared to those of the naïve hUCB-MSCs correlated with TLR-primed MSCs’ enhanced therapeutic efficacy against LPS-induced macrophages. This is consistent with our previous study, which confirmed the abolished therapeutic efficacy of VEGF-knockdown hUCB-MSC-derived EVs in the bronchopulmonary dysplasia animal model [57].

Consistent with our results, Ti et al. suggested a role for the EV-shuttled miRNA let-7b in TLR4 signaling in the hUCB-MSCs [41]. Multiple studies have emphasized the role of MSC-derived miRNAs in exerting their effects on immune cells via EVs [18,41]. The expression of different miRNAs is dependent on TLR activation in MSCs. Ti et al. confirmed that the expression level of miRNA let-7b was significantly higher in EVs isolated from TLR4-primed hUCB-MSCs than in those isolated from naïve hUCB-MSCs. They confirmed that let-7b inhibited the activation of TLR4 in macrophages by observing that isolated TLR4-primed EVs significantly induced M2 polarization of macrophages, increasing anti-inflammatory cytokines and reducing pro-inflammatory cytokines. This supports our finding that TLR4-primed EVs induce anti-inflammatory effects in LPS-induced macrophages. The parallel result from the indirect co-culture of TLR-primed hUCB-MSCs, using CMs and EVs, and LPS-induced macrophages, strongly suggests that the EV cargo plays a significant role in the immunomodulation of immune cells. EVs alone can exert an effect comparable to that of MSC treatment, and EVs are more clinically advantageous than MSCs considering their nature as cell-free vesicles, allowing them to be clinically flexible in dosing and modulation, and less tumorigenic [58,59,60]. Here, we suggest that in the LPS-induced macrophage in vitro model, EVs derived from both TLR3- and TLR4-primed MSCs could be potential therapeutic candidates for enhancing the polarization of macrophages towards the M2 phenotype.

This study investigated the TLR4-primed hUCB-MSCs’ anti-inflammatory and immunomodulatory properties against rodent macrophages, in parallel to our previously confirmed bronchopulmonary dysplasia and ALI in vivo studies. However, to further validate TLR3- and TLR4-primed hUCB-MSCs as clinical therapeutic candidates, the experiment must be repeated using human macrophages. We have not directly tested the effect of human-derived MSCs in non-stimulated resting rodent macrophages; however, there was no allogenic immune response observed in our previous in vivo studies confirmed by immunostaining of pan-macrophage marker ED1 and apoptotic analysis TUNEL staining and measurement of inflammatory cytokines in lung tissues. [10,26,61,62].

## 4. Materials and Methods

### 4.1. Preparation of hUCB-MSC

Human umbilical cord blood-derived MSCs (hUCB-MSCs) from a single donor were obtained from Medipost Co., Ltd. (Medipost Co., Ltd., Seoul, Republic of Korea) [63]. hUCB-MSCs were expanded and cultured as previously described under strict compliance with good manufacturing practice [64,65]. Isolated MSCs were confirmed to be CD73 and CD105 positive and CD14, CD34, and CD45 negative by fluorescence activated cell sorting (FACS). MSCs’ differential potentials and immunophenotypic results were analyzed as previously described [66,67]. No changes in the karyotype were observed at passage 11 [68]. hUCB-MSCs were cultured in α-MEM (Gibco, Grand Island, NY, USA) supplemented with 10% fetal bovine serum (Gibco, Grand Island, NY, USA) and 1% penicillin/streptomycin (Invitrogen, Carlsbad, CA, USA) in a humidified incubator with 5% CO_2_ at 36 °C. The culture media was changed every 2–3 days. hUCB-MSCs at passage 6 were used in this study.

### 4.2. Priming of hUCB-MSCs

When hUCB-MSCs reached approximately 90% confluency, culture media was replaced with a serum-free ɑ-MEM media supplemented with LPS (10 ng, 100 ng, 1 µg, 10 µg) or poly(I:C) (100 ng, 1 µg, 10 µg, 100 µg) and further cultured for 6 h to prime for TLR3 and TLR, respectively, following the previously established protocol [26]. LPS from *E. coli* O111:B4 (#L2630) and poly(I:C) (#P1530) were purchased from Sigma Aldrich (Sigma Aldrich, Burlington, MA, USA).

### 4.3. Preparation of hUCB-MSC Conditioned Media

hUCB-MSCs were seeded in a 6-well plate and expanded until they reached 90% confluency. Then, 100 µL of LPS, poly(I:C), or phosphate-buffered saline (PBS) for normal control was added to 2 mL of serum-free media per well. After 6 h, the conditioned medium was collected, filtered with a 0.2 µm pore-size syringe filter (Nalgene, Rochester, NY, USA), and stored at −80 °C until further use.

### 4.4. Primary Culture of Rat Alveolar Macrophages

The primary culture of rat AM was prepared using the method previously described by Engwall and Li [69]. In brief, bronchoalveolar lavage fluid (BAL) was obtained from 6–8 week-old SD rats, by washing rat lungs with PBS through the trachea. The obtained BAL fluid was centrifuged at 250× *g* for 5 min at 4 °C. Pelleted cells were resuspended in 1 mL of RPMI medium containing 10% FBS (Gibco, Grand Island, NY, USA) and 1% penicillin/streptomycin (Invitrogen, Carlsbad, CA, USA). The cells were seeded at 5 × 10^5^ cells/well in a 24-well culture plate (Falcon, Glendale, AZ, USA) with RPMI media, in a humidified incubator with 5% CO_2_ at 36 °C. The culture media was changed every 2–3 days.

### 4.5. In Vitro Model of LPS-Induced Inflammation in Alveolar Macrophages

The RAW 264.7 cell line was obtained from the Korean Cell Line Bank. RAW 264.7 cells and primary cultured rat AM were grown in RPMI supplemented with 10% FBS and 1% penicillin/streptomycin in a 24-well culture plate until they reached 90% confluency. At 90% confluency, cells were treated with MSCs, CM, or EV dissolved with 1µg/mL LPS (LPS O111:B4; Sigma Aldrich, Burlington, MA, USA) for 24 h. MSCs were co-cultured with macrophages in a 1:10 ratio using a 0.4 µm pore Transwell insert (Corning Inc., Corning, NY, USA). For CM treatment, 200 µL of CM was added to the macrophages. For EV treatment, 10 µL/mL of EVs was added to the cells.

### 4.6. EV Isolation and Quantification

Immediately after priming the hUCB-MSCs, the conditioned media were centrifuged at 3000 rpm for 30 min at 4 °C to remove cellular debris. The collected supernatant was further centrifuged at 100,000× *g* for 120 min at 4 °C to sediment the EVs. The supernatant was collected and stored at −80 °C as an EV-removed CM. The pellet was washed twice, resuspended in PBS, and stored at −80 °C until further use. The isolated EV was quantified by measuring the rate of Brownian motion using a NanoSight (NanoSight NS300, Malvern, Worcestershire, UK).

### 4.7. Primary Culture of Rat Alveolar Macrophages

Flow cytometric analysis was used to quantify the extent of macrophage polarization. RAW 264.7 cells and primary cultured rat AM were pelleted by centrifuging for 10 min at 450× *g* at 4 °C. For rat AM, RBC was removed from BAL fluid using RBC lysis buffer (Sigma Aldrich; Burlington, MA, USA) before pelleting the cells for FACS. Anti-CD86 and anti-CD204 were used to determine the extent of M1/M2 polarization of RAW 264.7 cells and rat AM [70]. All antibodies for flow cytometry were obtained from BioLegend (San Diego, CA, USA).

### 4.8. Enzyme-Linked Immunosorbent Assay

Pro-inflammatory cytokines IL-6, TNF-a, and IL-1α, and growth factors VEGF and HGF were measured from the TLR3- and TLR4-primed hUCB-MSC conditioned media using human Luminex^®^ Discovery Assay (R&D Systems, Minneapolis, MN, USA). Rat and mouse Luminex^®^ Discovery Assays were used to measure pro-inflammatory cytokines IL-6, TNF-a, and IL-1α from LPS-induced rat AM and RAW 264.7 cells, respectively (R&D Systems, Minneapolis, MN, USA). ELISA was performed following the manufacturer’s protocol.

### 4.9. Statistical Analysis

Data are presented as mean ± standard error of the mean (SEM). One-way analysis of variance (ANOVA) and Tukey’s post hoc test were used to statistically compare between groups. All data were analyzed using GraphPad Prism version 5.01 (GraphPad, San Diego, CA, USA). *p*-values less than 0.05 were considered statistically significant.

## 5. Conclusions

Our data showed that high doses and 6-h stimulation with TLR3 and TLR4 of hUCB-MSCs enhanced their anti-inflammatory and immunosuppressive effects against inflammatory macrophages. Priming TLR3 and TLR4 of hUCB-MSC enhanced the reduction of pro-inflammatory cytokines and M2 polarization of macrophages compared to that observed in naïve hUCB-MSCs against LPS-induced inflammation in an in vitro model. TLR3- and TLR4-primed MSC-derived EVs alone were capable of modulating macrophage polarization. The ability of EVs derived from both TLR3- and TLR4-priming hUCB-MSCs to modulate the macrophage phenotype suggests that TLR-primed EVs are potential therapeutic candidates for the treatment of *E. coli*-induced ALI in preclinical studies.

## Figures and Tables

**Figure 1 ijms-24-16264-f001:**
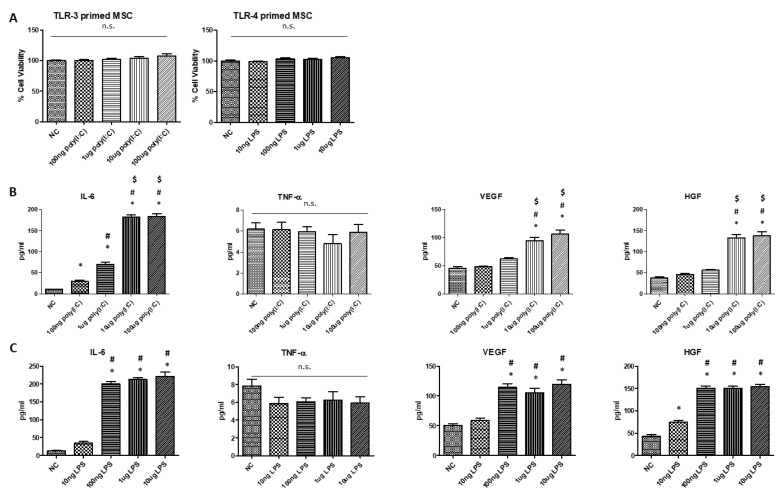
Dose-dependent effect of poly(I:C) and LPS on viability of hUCB-MSCs and their secretion of cytokines and growth factors. (**A**) Viability of hUCB-MSCs upon stimulation with increasing doses of poly(I:C) and LPS. (**B**) Measured IL-6, TNF-a, VEGF, and HGF levels in response to stimulation with increasing doses of poly(I:C) (*n* = 6 in all groups. *; *p* < 0.05 compared to NC. #; *p* < 0.05 compared to 100 ng. $; *p* < 0.05 compared to 1 μg). (**C**) Measured IL-6, TNF-a, VEGF, and HGF levels in response to stimulation with increasing doses of LPS (*n* = 6 in all groups. *; *p* < 0.05 compared to NC. #; *p* < 0.05 compared to 10 ng). NC: Normal Control. n.s.: not statistically significant. Data are shown as mean ± SEM. One-way ANOVA with Tukey’s post hoc test.

**Figure 2 ijms-24-16264-f002:**
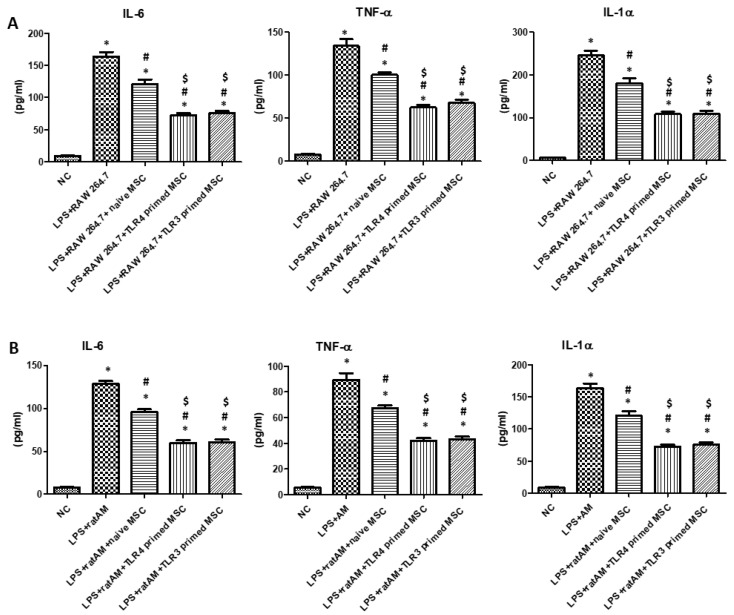
Effect of hUCB-MSC treatment on cytokine secretion of LPS-induced alveolar macrophages. (**A**) IL-6, TNF-a, and IL-1a levels measured in LPS-induced RAW264.7 cells and (**B**) LPS-induced rat AM. NC: normal control. MSC: hUCB-MSC. Rat AM: primary cultured rat alveolar macrophages. (*; *p* < 0.0001 vs. NC group. #; *p* < 0.0001 vs. LPS + RAW264.7. $; *p* < 0.0001 vs. LPS + RAW264.7 + naïve MSC). Data are shown as mean ± SEM. One-way ANOVA with Tukey’s post hoc test.

**Figure 3 ijms-24-16264-f003:**
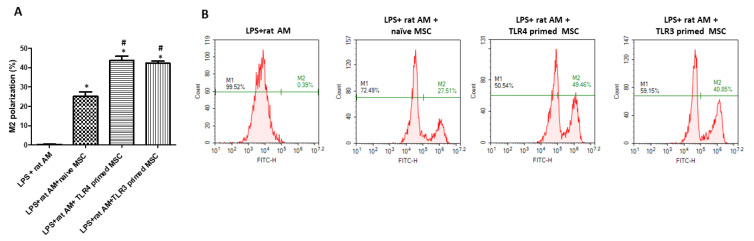
Percentage representation of M1 and M2 polarization in LPS-induced rat AM after MSC treatment. (**A**) Bar graph and (**B**) FACS histogram representation of the extent of M2 polarization in rat AM after treatment with naïve, TLR3-, and TLR4-primed hUCB-MSCs. NC: normal control. Rat AM: primary cultured rat alveolar macrophages (*n* = 4 in all groups. *; *p* < 0.0001 vs. LPS+ rat AM. #; *p* < 0.0001 vs. LPS+ rat AM+ naïve MSC). Data are shown as mean ± SEM. One-way ANOVA with Tukey’s post hoc test.

**Figure 4 ijms-24-16264-f004:**
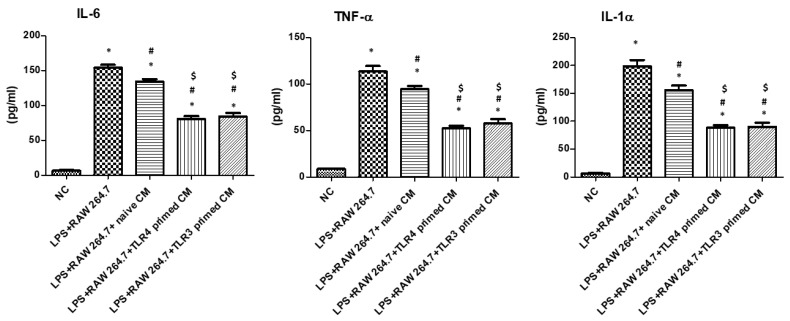
Measurement of pro-inflammatory cytokines in LPS-induced RAW264.7 cells after treatment with MSC-CMs. NC: normal control. CM: conditioned medium (*n* = 4 in all groups. *; *p* < 0.0001 vs. NC. #; *p* < 0.0001 vs. LPS + RAW264.7. $; *p* < 0.0001 vs. LPS + RAW264.7 + naïve CM). Data are shown as mean ± SEM. One-way ANOVA with Tukey’s post hoc test.

**Figure 5 ijms-24-16264-f005:**
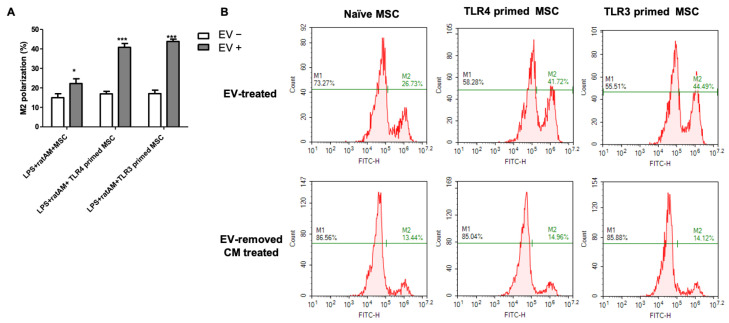
Percent representation of M2 polarization in LPS-induced rat AM after the treatment with MSC-derived EVs. (**A**) Bar graph and (**B**) FACS histogram representing the extent of M2 polarization in rat AM after treatment with isolated EVs or EV-removed CM from naïve, TLR3-, and TLR4-primed hUCB-MSCs. Rat AM: primary cultured rat alveolar macrophages (*n* = 4 in all groups. *; *p* < 0.05 ***; *p* < 0.0001 vs. EV-removed CM treated group). Data are shown as mean ± SEM. Two-way ANOVA with Bonferroni post hoc test.

## Data Availability

The data generated in this study are included in the paper.

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
