# Peer review of "Mesenchymal Stromal Cells Primed by Toll-like Receptors 3 and 4 Enhanced Anti-Inflammatory Effects against LPS-Induced Macrophages via Extracellular Vesicles"

_ijms, 2023, doi:10.3390/ijms242216264_

Round 1

Reviewer 1 Report

Comments and Suggestions for Authors

The manuscript “Mesenchymal stromal cells primed by Toll-like receptors 3 and 4 enhanced anti-inflammatory effects against LPS-induced macrophages via extracellular vesicles” deals with the human umbilical cord blood mesenchymal stromal cells primed by Toll-like receptors 3 and 4 respond an anti-inflammatory phenotype mediated by extracellular vesicles.

The topic discussed is very important in the treatment and prevention of inflammation.

I would like to make a few comments:

1)        Figure 2: There is no data of the variant with naïve MSCs without LPS. In the article hUCB-MSCs were used with rat alveolar macrophages and mouse cell line. It is necessary to use the second negative control: hUCB-MSCs with rat alveolar macrophages and hUCB-MSCs with RAW264.7 to prove that there was no action of human hUCB-MSCs on rodent cells. The same comment is to the figure 4.

2)        Lines 346-347: Indicate please the type and manufacturer of LPS and poly(I:C).

3)        Line 397 “4.7 Enzyme-Linked Immunosorbent Assay”:

Comment:  It is necessary to mention the type and manufacturer of ELISA kit for IL-6, TNF-a, VEGF, and HGF in hUCB-MSCs. There is no information in the item “Materials and methods” about VEGF and HGF measuring. If two types of ELISA kits were used (for mouse and human cytokines), it should be described.

4)        When you are writing about TLRs in the item “Discussion” please discuss the complex nature of innate immune receptors regulation inflammation. In particular, prolonged stimulation with LPS can reduce the severity of the allergic inflammation. Please discuss and cite the article:

Guryanova, S.V.; Gigani, O.B.; Gudima, G.O.; Kataeva, A.M.; Kolesnikova, N.V. Dual Effect of Low Molecular Weight Bioregulators of Bacterial Origin in Experimental Model of Asthma. Life 2022, 12, 192. https://doi.org/10.3390/life12020192

5)        When you are writing about anti-inflammatory activity of MSCs in the item “Discussion” please discuss mention the articles:

Salari, V.; Mengoni, F.; Del Gallo, F.; Bertini, G.; Fabene, P.F. The Anti-Inflammatory Properties of Mesenchymal Stem Cells in Epilepsy: Possible Treatments and Future Perspectives. Int. J. Mol. Sci. 2020, 21, 9683. https://doi.org/10.3390/ijms21249683

Jiang B, Fu X, Yan L, Li S, Zhao D, Wang X, Duan Y, Yan Y, Li E, Wu K, Inglis BM, Ji W, Xu RH, Si W. Transplantation of human ESC-derived mesenchymal stem cell spheroids ameliorates spontaneous osteoarthritis in rhesus macaques. Theranostics. 2019 Aug 21;9(22):6587-6600. doi: 10.7150/thno.35391.

6)        It should be mentioned the limitation of the investigation: for example, the usage rodent macrophages and rodent cell line. In future it is necessary to explore the influence of extracellular vesicles from human mesenchymal stromal cells on the human macrophages.

Reviewer 2 Report

Comments and Suggestions for Authors

The study of the anti-inflammatory activity and extracellular vesicle-mediated immunomodulatory effect of HUCB-MSCs in an LPS-induced inflammatory microenvironment is of great importance and highly topical.

I think the introduction would benefit from a paragraph on the factors that are modulated during inflammation, which would shed more light on your forthcoming results. 

In the results section, it would be good to specify that it's about cytokine secretion. For example, in figure 2, and for all other figures where necessary. 

Protocol: Can you justify the 6H time?
